# The N-terminal domain of RfaH plays an active role in protein fold-switching

Pablo Galaz-Davison[1,2], Ernesto A. Román[3,4], César A. Ramírez-Sarmiento[1,2]*

**1** Institute for Biological and Medical Engineering, Schools of Engineering, Medicine and Biological Sciences, Pontificia Universidad Católica de Chile, Santiago, Chile, **2** ANID–Millennium Science Initiative Program–Millennium Institute for Integrative Biology (iBio), Santiago, Chile, **3** Instituto de Química y Fisicoquímica Biológicas (UBA-CONICET), Ciudad Autónoma de Buenos Aires, Argentina, **4** Facultad de Ciencias Exactas y Naturales, Universidad de Buenos Aires, Ciudad Autónoma de Buenos Aires, Argentina

* cesar.ramirez@uc.cl

**Data Availability Statement:** All produced trajectories, representative structures after refolding for all simulation systems, data analysis,

## Abstract

The bacterial elongation factor RfaH promotes the expression of virulence factors by specifically binding to RNA polymerases (RNAP) paused at a DNA signal. This behavior is unlike that of its paralog NusG, the major representative of the protein family to which RfaH belongs. Both proteins have an N-terminal domain (NTD) bearing an RNAP binding site, yet NusG C-terminal domain (CTD) is folded as a β-barrel while RfaH CTD is forming an α-hairpin blocking such site. Upon recognition of the specific DNA exposed by RNAP, RfaH is activated via interdomain dissociation and complete CTD structural rearrangement into a β-barrel structurally identical to NusG CTD. Although RfaH transformation has been extensively characterized computationally, little attention has been given to the role of the NTD in the fold-switching process, as its structure remains unchanged. Here, we used Associative Water-mediated Structure and Energy Model (AWSEM) molecular dynamics to characterize the transformation of RfaH, spotlighting the sequence-dependent effects of NTD on CTD fold stabilization. Umbrella sampling simulations guided by native contacts recapitulate the thermodynamic equilibrium experimentally observed for RfaH and its isolated CTD. Temperature refolding simulations of full-length RfaH show a high success towards α-folded CTD, whereas the NTD interferes with βCTD folding, becoming trapped in a β-barrel intermediate. Meanwhile, NusG CTD refolding is unaffected by the presence of RfaH NTD, showing that these NTD-CTD interactions are encoded in RfaH sequence. Altogether, these results suggest that the NTD of RfaH favors the α-folded RfaH by specifically orienting the αCTD upon interdomain binding and by favoring β-barrel rupture into an intermediate from which fold-switching proceeds.

## Author summary

Proteins commonly adopt a single three-dimensional structure that is required for biological function. Nevertheless, proteins are not isolated in the cell, and the presence of binding partners can give rise to alternate structural configurations. Metamorphic proteins represent an extreme case of the latter, by folding into at least two well-defined

and the AWSEM and LAMMPS version used for these simulations, are available for download at the laboratory's simulation archive in the Open Science Framework (OSF, https://osf.io/bn6u3/).

**Funding:** This research was funded by the National Agency for Research and Development (ANID) FONDECYT Regular Grant 1201684 (to CARS) and ANID Millennium Science Initiative Program ICN17_022. PGD was supported by ANID Doctoral Scholarship 21181705. EAR has a research grant from Agencia Nacional de Promoción Científica y Tecnológica (PICT 2016-0014). Powered@NLHPC: This research was partially supported by the supercomputing infrastructure of the NLHPC (ECM-02). The funders had no role in study design, data collection and analysis, decision to publish, or preparation of the manuscript.

**Competing interests:** The authors have declared that no competing interests exist.

configurations that are both structurally and functionally different. For RfaH, a virulence factor in enterobacteria, two distinct folds are found: an autoinhibited state in which its two protein domains strongly interact, and an active state in which these domains dissociate due to a specific DNA signal on RNA polymerases. This activation is accompanied by the refolding of the C-terminal domain (CTD) from an α-helical structure to a β-barrel. Our work employs computational simulations to explore the role of the N-terminal domain (NTD) in regulating the metamorphic behavior of RfaH, determining that this domain has a major part in orienting and binding to the CTD in its α-helical fold, and in stabilizing an intermediate state instead of the fully folded β-barrel. These results suggest that the NTD not only participates in stabilizing the autoinhibited state, but also aids in fold-switching back to it after active RfaH is released from RNA polymerase.

## Introduction

The NusG/Spt5 family of transcription regulators is universally conserved in all three domains of life. *E. coli* NusG displays two domains in its structure, named N-terminal (NTD) and C-terminal domains (CTD) due to their location in the sequence [1]. The NTD is structurally conserved, folding as an α/β sandwich containing an hydrophobic depression that serves as binding site for the RNA polymerase (RNAP) [2], whereas the CTD folds as a small β-barrel that recruits the ribosome for coupled transcription-translation as well as other partners that regulate transcription (Fig 1) [3–5].

The elongation factor RfaH of *E. coli* is a clear outlier of the NusG family of transcription factors, having an NTD with the canonical protein family structure but a CTD that is folded as an α-helical hairpin rather than the classical β-barrel [6]. This conformation makes up the autoinhibited state of RfaH, as the α-folded CTD is blocking the RNAP binding site located at the NTD and impedes the spontaneous binding to the transcription elongation complex (TEC), i.e. the RNA polymerase in complex with DNA and RNA [6]. This autoinhibition is relieved when the transcribing polymerase pauses at a DNA sequence named *operon polarity suppressor* (*ops*) [7], whose exposed non-template strand forms a DNA hairpin acting as a recruiting partner for RfaH to the RNA polymerase [8–10], promoting interdomain dissociation and NTD binding to the β and β' subunit of RNAP [11,12]. Strikingly, the dissociated CTD refolds from the initial α-hairpin to a canonical β-barrel which serves as recruiting partner to the ribosomal protein S10, coupling transcription with translation (Fig 1) [3,11,13].

Numerous studies have addressed the metamorphosis of RfaH through a computational approach, in part due to the difficulties of observing the process in solution since the trigger for RfaH interdomain dissociation is the entire TEC. There have been reports indicating the possible pathways through which the isolated CTD may refold from the α- to the β-fold [14–17], which differ from the ones proposed when the CTD is accompanied by the NTD [18–20]. These results suggest that interactions formed between both domains strongly aid in stabilizing the α-fold as well as forming intermediate states that enable the transition between folds [18]. Nevertheless, these studies have focused mostly on the CTD transformation, leaving aside the details of how the NTD stabilizes the α-fold or its effects over the β-folded CTD after release from the TEC. The specifics of NTD-induced energetics on RfaH are not trivial, since the structure of RfaH-NTD [10] displays a more hydrophobic patch than that of NusG [11,21], which has been simultaneously associated to a tighter binding to RNAP, being RfaH NTD the only trigger required for fold-switching back from the active into the autoinhibited state [22].

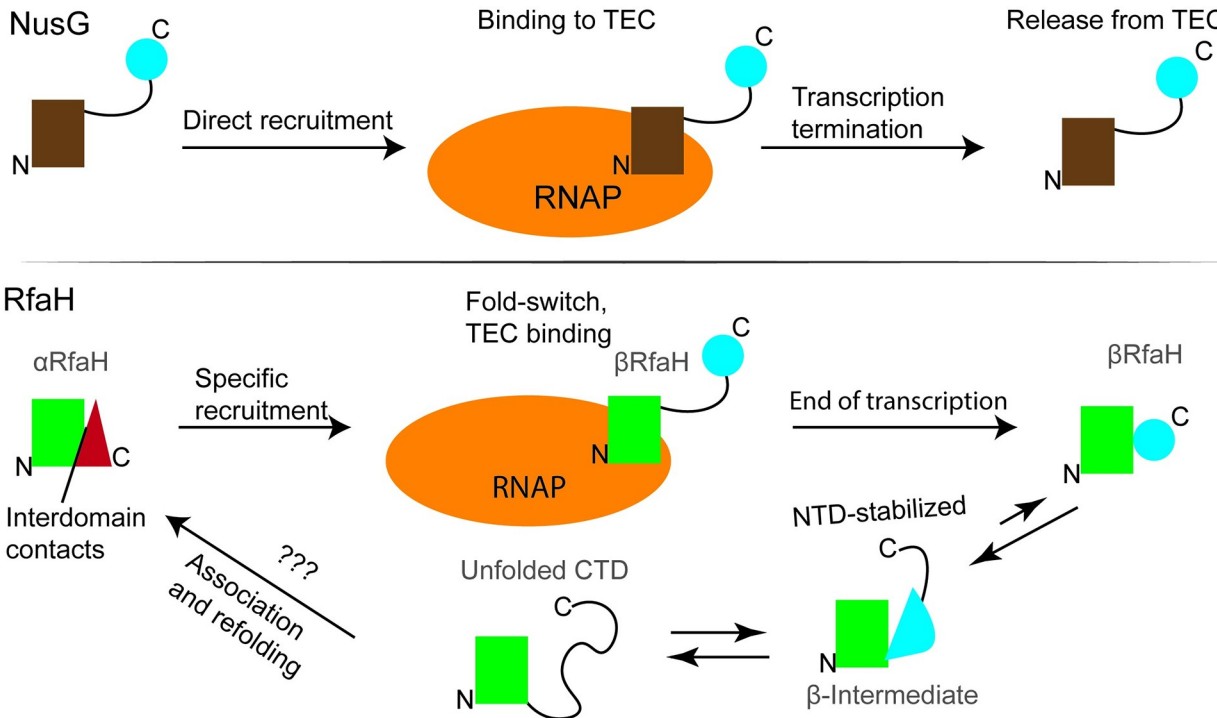

**Fig 1.** Schematic representation of the folding states of NusG (top) and RfaH (bottom) upon binding to and release from the transcription elongation complex (TEC). For RfaH a fold-switch is involved in this process, in which the steps after release from the TEC corresponding to partial unfolding into a β-intermediate and transiting the unfolding state before refolding into the autoinhibited state are based on the results presented in this article.

In this work, we relied on the Associative Water-Mediated Structure and Energy Model (AWSEM) to determine the effect that the NTD of RfaH has on the overall transformation energetics and the configurational space of both folds. AWSEM is a transferable force field, coarse-grained to three beads per residue ($C_\alpha$, $C_\beta$ and O), initially used to predict protein structure [23]. As a force field, it has been successfully used to study the NF-κB/IκB/DNA regulatory system [24], the nucleosome dynamics and energetics [25] and to determine the energy landscape of aggregation of the amyloid-β protein [26], among others. Unlike common atomistic force-fields, its energy potentials and granularity have been developed for efficiently explore protein folding while robustly carrying enough information to represent up to the dihedral behavior of the main chain. This is a significant step up from our previous works on RfaH using a structure-based $C_\alpha$ model [18], as not only we are now reducing the granularity but also increasing the roughness of the energy surface by including potentials for hydrogen bonding and solvent exposure propensity of each residue as well as residue-residue pairwise potentials that consider residue identity [23].

Using umbrella sampling, we determined the change in stability associated to interdomain separation and subsequent fold-switching, recapitulating the experimentally determined equilibrium of the system. That is, RfaH is much more stable in the α-configuration, but the β-folded CTD becomes much more stable in the absence of the NTD. Further temperature refolding simulations in the absence of information of known interdomain contacts showed that the highly hydrophobic side of the α-folded CTD consistently looks for an interaction partner and the NTD provides a suitable surface for its stabilization, recapitulating the binding orientation experimentally observed in solved structures of the autoinhibited state of RfaH. At the same time, the NTD interferes with βCTD refolding by mostly trapping it into a β-barrel intermediate, which is also observed in its metamorphic pathway. Altogether, these results

suggest that the NTD favors the CTD transformation towards the α-folded CTD by simultaneously stabilizing the α-hairpin and switching the equilibrium to favor β-barrel rupture into a β-intermediate state that is part of the refolding pathway towards its autoinhibited state.

## Methods

### Initial structures for molecular dynamics

The structure of the full-length RfaH protein in its α-state (αRfaH hereafter) was extracted from the crystal structure deposited in the Protein Data Bank (PDB) with accession ID 5ond, and so was the α-folded CTD (αCTD hereafter). The isolated β-folded CTD (βCTD hereafter) was extracted from the first NMR solution model of the PDB accession ID 2lcl, whereas the full-length version of the active β-folded RfaH (βRfaH hereafter) was extracted from the cryoEM RfaH:TEC structure with PDB accession ID 6c6s. On the other hand, the isolated CTD of NusG was extracted from the first model of the NMR-determined structure with PDB accession ID 2jvv.

### The AWSEM force field

The Associative Water-mediated Structure and Energy Model, AWSEM, [23] is a coarse-grained molecular dynamics (MD) protein folding model implemented in LAMMPS [27]. The granularity and efficiency of this model is achieved by reducing the number of atoms per residue to three beads, the $C_\alpha$, $C_\beta$ and O atoms, with the rest of them being calculated from ideal backbone geometry. This model contains five energy terms, which are extensively described in the work by Davtyan and cols [23] and are briefly summarized below:

$$V_{total} = V_{backbone} + V_{contact} + V_{burial} + V_{hydrogen\ bonding} + V_{memory} \qquad (1)$$

Of these terms, the *backbone* energy term guides the atoms to a protein-like geometry, which is achieved using potentials that ensure atom connectivity, chirality, Ramachandran distribution, and excluded volume interaction. The *contact* term defines $C_\beta$-$C_\beta$ distances and is responsible for the formation of residue-residue interactions in an amino acid-dependent manner. This potential includes pairwise direct contact potentials and many-body water-mediated contact potentials. The *burial* energy term is a many-body interaction potential that regulates solvent exposure of the protein core, depending on whether a residue has propensity to be in a low, medium or high-density environment. The *hydrogen bonding* term replicates the contacts of carbonyl oxygen to amide nitrogen formed in α-helices, parallel β-sheets and anti-parallel β-sheets. This potential includes additive terms for hydrogen bonding and cooperative stabilization terms for β-sheets, which we modified such that sheets of a minimum length of 3 residues can form, as the shortest strands observed in the β-barrels of NusG and RfaH are of this length. Finally, the *memory* term is a local bias applied to overlapping fragments from 3 to 9 residues that guides $C_\alpha$ and $C_\beta$ distances to those of a reference structure, being the only native bias that is used in these simulations. This potential has the form:

$$V_{memory} = -\lambda_{memory} \sum_m \omega_m \sum_{ij} e^{-\frac{(r_{ij}-r_{ij}^m)^2}{2\sigma_{ij}^2}} \; ; \; \text{with } \sigma_{ij} = |j-i|^{0.15} \text{Å} \qquad (2)$$

In this equation, the outer sum is carried out over all the aligned memory fragments, i.e., all short overlapping segments that share high sequence identity to a library of known proteins structures, with $\omega_m$ corresponding to the memory weight. The inner sum is carried out over the $C_\alpha$ and $C_\beta$ *i,j* pairs that are separated by at least 2 residues, with $r_{ij}$ being the distance between the atoms and $r_{ij}^m$ the distance in the reference fragment. Finally, $\lambda_{memory}$ corresponds to a scaling factor of the strength of this potential relative to the other terms. This potential can be guided to

multiple structures in a simulation, or as used in this work, limited to a single or two reference structures [23]. Also, the $\lambda_{memory}$ used in this work is of 0.3 compared to the default value of 0.2, resulting in a higher cooperativity due to a decrease in the roughness of the final potential.

## Calculation of $Q_{diff}$ and umbrella sampling

Normally, MD simulations sample configurations that are very close to the initial structure, hence observing structural transitions such as RfaH fold-switching would be a rare event that would require a very long simulation time. A way to overcome this is by using enhanced sampling strategies, such as umbrella sampling. This technique enables exploring poorly sampled regions of the configurational space by applying an external bias along a reaction coordinate that describes the transition between both RfaH folds. Generally, this external bias corresponds to a harmonic potential that is applied to multiple different reaction coordinate values, such that different simulations thoroughly sample a narrow phase space while ensuring the potential energy overlap between simulations at adjacent values along the reaction coordinate. The potential energy and reaction coordinate values from multiple independent simulations are then used as input for the Weighted Histogram Analysis Method (WHAM) [28] that returns the unbiased free energy landscape of RfaH fold-switching.

For the umbrella sampling method, 51 simulations of $2.4 \cdot 10^7$ timesteps or 120 ns each were run, and energy and frames were collected every 1,000 timesteps. The initial configuration was that of the unfolded isolated CTD and a dual memory approach was used, i.e., the fragments were driven to the memory of $\alpha$CTD and $\beta$CTD with equal strength. Similarly, for the full-length protein the initial state was that of the folded NTD plus unfolded CTD. The simulations sampled fractions of an order parameter called $Q_{diff}$ which corresponds to [26,29]:

$$Q_{diff} = \frac{q - q_A}{q_B - q_A}; \; where \tag{3}$$

$$q = \frac{1}{(N-2)(N-3)} \sum_{j>i+2} [e^{-(r_{ij}-r_{ij}^A)^2/2\sigma_{ij}^2} + 1 - e^{-(r_{ij}-r_{ij}^B)^2/2\sigma_{ij}^2}]; \; with \; \sigma_{ij} = |j-i|^{0.15} \text{Å} \tag{4}$$

Where $N$ is the sequence length, $q_A$ and $q_B$ are constants obtained by evaluating the $q$ function in the two structures to which the transition is to be interpolated, and $r_{ij}$ measures the $C_\alpha$ distance between residue $i$ and $j$ in the simulation, where superscripts A and B refer to such distance in each reference structure. This is evaluated for all contacts between $j>i+2$ residues whose $C_\alpha$ are at 9.5 Å or below in at least one of the reference structures. This distance is calculated from a C$\alpha$-C$\alpha$ distance matrix for RfaH (S1 Fig) or the isolated CTD. In the case of the full-length protein, the NTD was excluded from the calculations for the autoinhibited and active RfaH configurations, as it does not experience a conformational change during RfaH activation. Interdomain contacts in the starting structure for $\beta$RfaH were also excluded. These exclusions were achieved by increasing the residue-residue distances within the NTD and between the NTD and CTD of $\beta$RfaH in the distance matrices to 99 Å.

Using the $Q_{diff}$ value, a bias is applied by adding a new potential to the system with the form:

$$V_{Umbrella} = \frac{1}{2} k (Q_{diff} - Q_0)^2 \tag{5}$$

Where $k$ is the harmonic potential constant, here 1,500 kcal·mol$^{-1}$, and $Q_0$ being the center of the distribution of a $Q_{diff}$ value ranging from 0.00 to 1.00 by increments of 0.02. From these simulations the potential energy and $Q_{diff}$ values were obtained for each frame, as well as the

$C_\alpha$ RMSD of the best-fit against both reference CTD folds that were calculated using VMD [30]. The simulations exploring the same $Q_{diff}$ range were run at two temperatures, 650 K and 750 K, and the AWSEM temperature units were expressed as folding temperature ($T_f$) by expressing these temperatures relative to the folding temperature of full-length αRfaH (~650 K). Histograms of these quantities show overlap between simulations at adjacent $Q_{diff}$ values (S2 Fig). The RMSD against αCTD and βCTD were then used as reaction coordinates for thermodynamic analysis using the WHAM algorithm [31] implemented in Java [32]. For this analysis, the first 4,000 frames or 20 ns were excluded as this was the equilibration time from the unfolded state to the desired biased configuration.

## Refolding simulations

For these simulations, random initial unfolded configurations for each system were generated by running 100,000 timesteps of 5 fs of a simulation without any potential but the backbone energy term, saving a simulation restart configuration every 10,000 timesteps. The restart configuration with the lowest $Q_W$ value, which in all cases was below 0.1, was used as a starting configuration for the refolding simulations. All 100 simulations were randomly assigned initial velocities and run for $3 \cdot 10^7$ timesteps of 5 fs, totaling 150 ns each, during which the temperature linearly decreased from $1.5T_f$ to $0.6T_f$, where the temperature is expressed relative to the folding temperature ($T_f$) of αRfaH (S3 Fig), the predominant state in solution for full-length RfaH. All constructs were completely unfolded at the initial temperature and either completely refolded, trapped into an intermediate state or misfolded at the final temperature. The final structures of these simulations were clustered by calculating pairwise best-fit RMSD [33] using Chimera [34]. For the representative member of each cluster, as well as for non-clustered models, the secondary structure assignment was calculated using STRIDE [35]. These secondary structure assignments are summarized in S1 Table alongside the corresponding $Q_W$, which is a measure of structural similarity to a given structure and obtained using the formula [36]:

$$Q_w = \frac{2}{(N-2)(N-3)} \sum_{j>i+2} e^{-(r_{ij}-r_{ij}^N)^2/2\sigma_{ij}^2}; \ with \ \sigma_{ij} = |j-i|^{0.15} \text{Å} \tag{6}$$

Where, similarly to $Q_{diff}$, $r_{ij}$ measures the $C_\alpha$ distance between residues $i$ and $j$ for the current and reference (superscript N) structure, given that the distance in the latter is lower than 9.5 Å, and $N$ stands for the number of residues in the protein.

## Results

### MD simulations of RfaH and its isolated CTD recapitulate their experimental states

The simplest question that can be asked to an energy model about RfaH is whether it can replicate the experimentally observed CTD populations of α and β folds. More precisely, the strong predominance of αRfaH for the full-length protein in solution [6,10], and of βCTD when this domain is isolated as the result of the NTD-CTD linker being cleaved or by purifying only this domain in solution [3].

To explore this scenario, we set up umbrella sampling simulations that guide the transformation of RfaH for two systems: one in which we modeled the transition in the context of the full-length protein, that is αRfaH and βRfaH, and another in which only its CTD is modeled transitioning between αCTD and βCTD. Specifically, 51 umbrella simulations were generated for each system at two temperatures, 1.0 and $1.15T_f$, where each simulation is energetically biased to explore a fraction of the configurations determined by a reaction coordinate named

$Q_{diff}$, resulting in a gradual exploration of the configurational space between the α- and β-states of either full-length RfaH or its isolated CTD.

This exploration of the transformation was then analyzed using WHAM [31], and the heat capacity was visually inspected (S3 Fig). To evaluate the change in stability between RfaH folds, free energy surfaces were calculated at a temperature just below the first peak in heat capacity for each system to ascertain the preferred folded state (Fig 2A and 2B).

The isolated CTD free energy surface displays two minima of similar free energy at low RMSD of βCTD and a higher free energy minimum at low RMSD of αCTD (Fig 2A). This suggests that the isolated CTD (residues 100 to 162) exist predominantly as a β-barrel, and it needs to cross an energy barrier of over 50 kcal·mol$^{-1}$ to reach the α-folded state. On the other hand, the energy landscape of the CTD in the context of the full-length protein displays a major free energy minimum that expands between 1 and 4 Å in RMSD to αCTD (Fig 2B), indicating that RfaH exists predominantly in the autoinhibited state. These results are consistent with the experimental evidence for full-length RfaH and the isolated CTD in solution [3].

The fold-switching path explored in our simulations is best observed when projecting the free energy surface onto coordinates that directly measure the structural transition of RfaH CTD, such as $Q_{diff}$ and the difference in RMSD between βCTD and αCTD. These transitions, shown in Fig 2C, were obtained at temperatures where the peak in heat capacity is observed for each system (S3 Fig).

In the case of the isolated CTD, the first peak in heat capacity is observed at $0.95T_f$ and corresponds to the transition between the folded βCTD and a folding intermediate. Meanwhile, a second peak in heat capacity is observed at $1.15T_f$ and corresponds to the transition between the β-intermediate and the unfolded state. In the first of these landscapes, the αCTD minimum is shown as a high and broad free energy minimum similarly to its basin observed in Fig 2B, a characteristic that likely arises from the structuredness of the helices, which have been ascertained in both simulations [18,37] and experiments [38].

By projecting the free energy into a single coordinate, namely $Q_{diff}$, the energy barriers involved in the fold-switching process can be observed more clearly (S4 Fig). The transition between the β-barrel and β-intermediate has an estimated free energy barrier of 6.4 kcal/mol, whereas the transition between the β-intermediate and the unfolded minimum has a free energy barrier of mere 1.5 kcal/mol. At $1.15T_f$ only the β-intermediate and unfolded states are observed, while at $0.95T_f$ the transition to the αCTD is better observed, separated by a free energy barrier of 30 kcal/mol with a transition state sitting at unfolded configurations.

In the free energy surfaces for the full-length RfaH protein only one transition is observed at its $T_f$. By analyzing its free energy barriers, it can be noted that a transition occurs between $Q_{dif}$ 0.7 and 0.9, with a barrier of 4.4 kcal/mol. Closer inspection of the structural characteristics of this second minimum show that it has a RMSD of around 2.5 Å to αCTD, indicating that the cooperative decrease in $Q_{diff}$ is explained by the dissociation and partial rupture of the αCTD. The second energy barrier observed separates the folded state from the unfolded configurations and has a similar energy of 4.6 kcal/mol. The free energy basin for βRfaH is not observed at this temperature.

Altogether, these results recapitulate the experimentally predominant folded state for each simulation system in solution, which is separated by a significant energy gap from their alternative native states. Our results also show that both folded states of RfaH are connected by the unfolded state as well as by a hypothetical three-strand intermediate observed in the simulations for the isolated CTD, thus proposing the following fold-switching mechanism:

$$\alpha RfaH \rightleftarrows Unfolded \rightleftarrows \beta-intermediate \rightleftarrows \beta RfaH$$

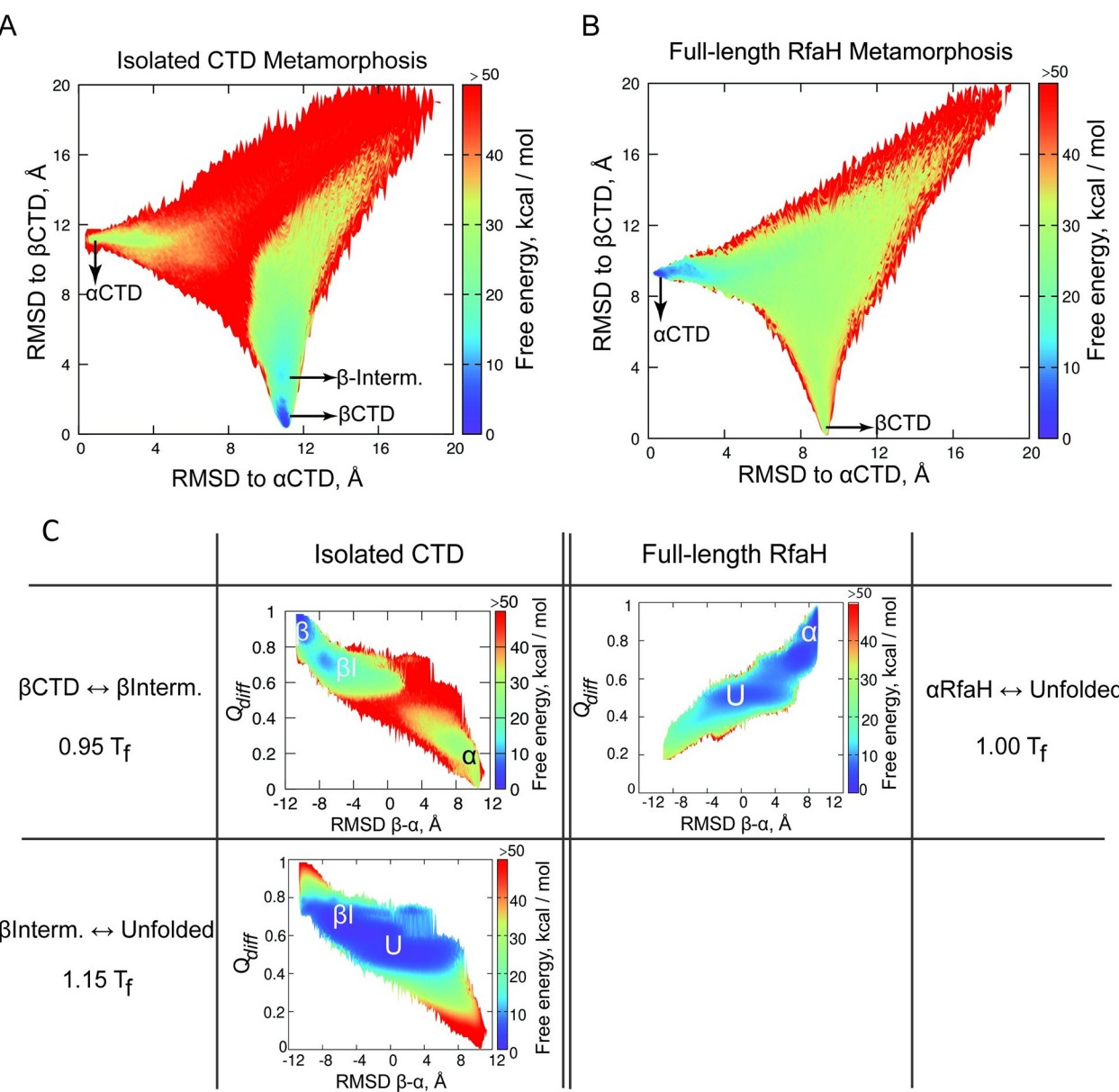

**Fig 2. Energetics of RfaH transformation.** (A, B) Free energy surface for the transformation of RfaH CTD in the full-length protein (A) or the isolated domain (B). The RMSD against the experimental αCTD and βCTD were used as reaction coordinates. (C) Free energy surface of the transitions of RfaH CTD in the context of the full-length protein with folded NTD or the isolated CTD, projected onto the transformation reaction coordinates $Q_{diff}$ and RMSD β-α. Here, βI corresponds to a folding intermediate, and U corresponds to the unfolded state.

## The NTD of RfaH strongly stabilizes the α-fold and hinders proper βRfaH refolding

One disadvantage of the umbrella sampling simulations is that it directly employs the number of native contacts of the system in αRfaH and βRfaH as collective variables to drive the structural interconversion of RfaH. Then, it becomes difficult to calculate the likelihood of other configurations that, albeit having a significant number of native contacts, may also display an important number of non-native contacts that could be relevant for its stabilization.

Consequently, one is unable to directly evaluate, for example, how the appropriate binding configuration between the NTD and CTD is guided by sequence features in RfaH.

AWSEM allows to restrict the use of structural biases only towards local-in-sequence interactions by using the fragment memory potential that limits the configurational exploration of short segments of the protein to those of a reference structure [23]. By not providing information about contacts between the NTD and CTD, these simulations freely explore the interdomain interaction landscape. A similar simulation strategy has been previously employed to correctly predict binding interfaces of both homodimers and heterodimers [39].

Using a temperature gradient through long MD simulations ($3 \cdot 10^7$ timesteps of 5 fs, compared to previously reported folding annealing simulations of $4 \cdot 10^6$ timesteps [23] and $6 \cdot 10^6$ timesteps [40]), 100 models with fragment memory to a single reference structure were allowed to refold starting from random unfolded conformations ($Q_W < 0.1$). In these single-memory models, only the NTD and CTD of RfaH, but not the linker connecting both domains, were given memory, and these memories are withdrawn from a single reference structure, either αRfaH or βRfaH. This approach leaves the linker that connects both domains with a major conformational freedom and results in the C- and N-terminal domains being structurally uncoupled, as the 10-residue long connector that exist between them is not part of the structural bias and therefore disrupts memory continuity. Therefore, any interdomain interaction formed in these simulations is the result of stabilizing residue-residue contacts encoded by the transferable part of the AWSEM force field, and not due to fragment memory or any other external potentials to favor its exploration. Using this approach, we simulated the refolding of αRfaH and calculated the amount of native tertiary contacts reached at the end of the simulation (Fig 3 and S1 Table).

Refolding simulations employing αRfaH as the single memory reference structure show that 81% of the trajectories reach the native state ($Q_W = 0.75$, Fig 3A). These predicted structures are characterized by the proper orientation and binding of the αCTD against the NTD (Fig 3B), recapitulating the experimentally solved structure of RfaH in its autoinhibited state [6], and is compatible with the observation that the full-length protein successfully refolds to

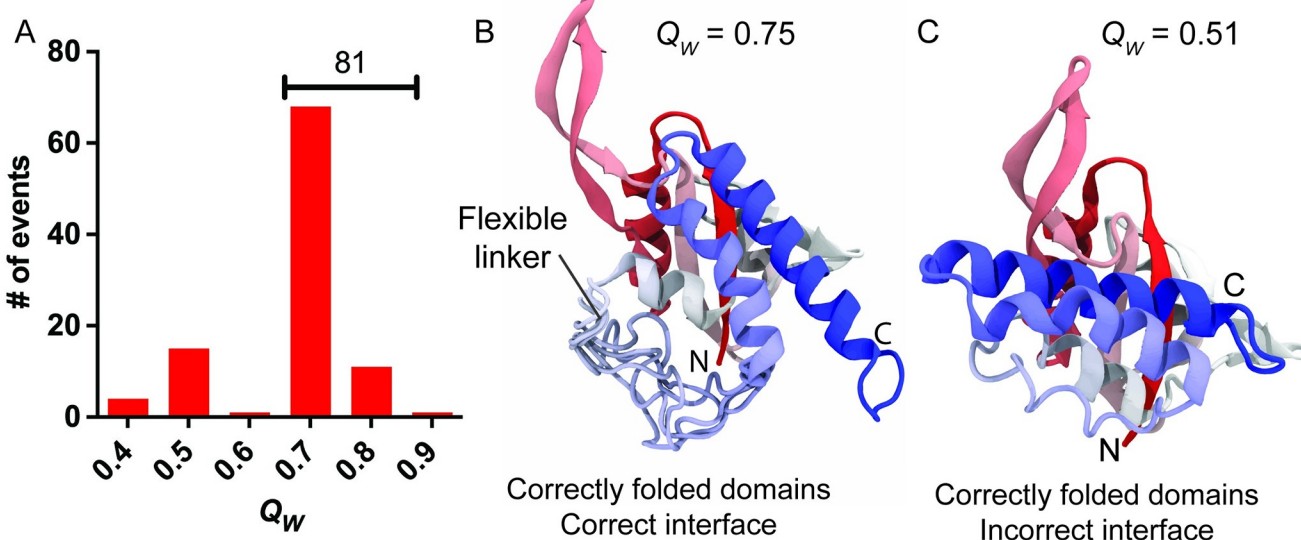

**Fig 3. Refolding efficiency of αRfaH.** (A) Distribution of tertiary contacts ($Q_W$) in the final structure of the 100 refolding simulations generated for αRfaH using a single memory. (B-C) Representative final structures after αRfaH refolding with high (B) and low (C) $Q_W$ respectively. The images are colored in gradient from red (N-terminus) to blue (C-terminus).

this state on its own [22]. This specificity is achieved despite the lack of structural biases on the interdomain interface and linker regions, and thus a result of sequence determinants in both the NTD and CTD of RfaH encoding this behavior. In fact, the linker is not stabilized in a particular conformation (Fig 3B) and does not form stable contacts with any domain. In all other trajectories the interdomain interface is formed incorrectly, although both the NTD and αCTD reach their native conformations mostly due to the fragment memory bias. Observation of the refolding traces (S5 Fig) show that the αCTD is only stabilized upon or after NTD folding, suggesting that the NTD is responsible for the stabilization and orientation of the αCTD.

To further assess the effect of the NTD hydrophobic patch in CTD folding, the same refolding experiment was performed for βRfaH extracted from the cryo-EM structure. To enlighten the effect that the NTD could have on βCTD refolding, the resulting structures are compared with equivalent refolding of the solved structure of the isolated βCTD. The results of βRfaH and βCTD refolding experiments are summarized in Fig 4 and S1 Table.

For the isolated domain, the βCTD refolds with a similar efficiency than αRfaH (75%), with the remainder of the simulations reaching an intermediate state characterized by a lower $Q_W$, in which only the three larger β-strands of the barrel are folded (Fig 4B). In stark contrast, the presence of RfaH NTD reduces the CTD refolding efficiency to only 29%, whereas all other refolding trajectories become trapped in the same β-intermediate observed for the isolated βCTD. These results suggest that the stabilization of this intermediate is a result of specific NTD-CTD interactions established during the folding process of βRfaH.

To determine that the βCTD intermediate is stabilized by specific interactions between both RfaH domains, a harmonic potential was used to maintain the NTD and CTD domains away from each other during refolding simulations of βRfaH. Upon keeping both domains

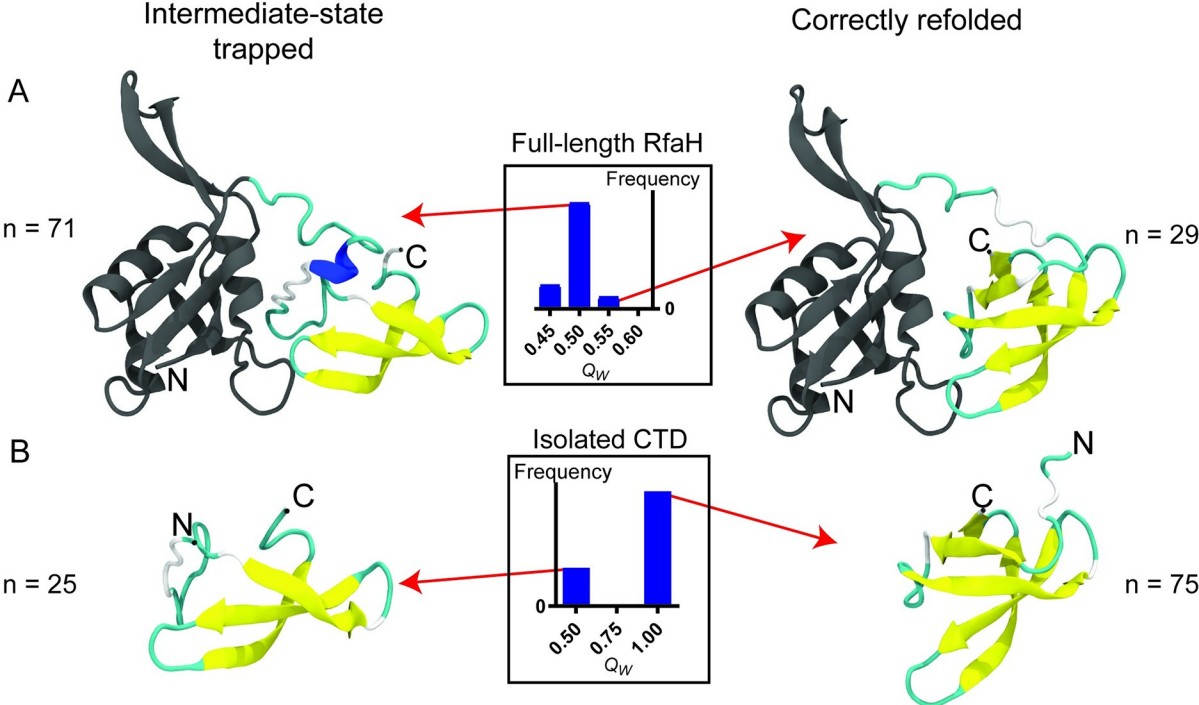

**Fig 4. Refolding of βCTD in the context of the full-length protein and in isolation.** Representative final structures after βCTD refolding in the context of the full-length protein (A) and in isolation (B). The histograms represent the $Q_W$ distribution of the final structures. The intermediate state is formed by the three largest β-strands that form the CTD barrel, namely strands β2, β3 and β4.

apart throughout the simulation, the βCTD mostly refolds as if it was isolated, with 66% of cases achieving complete refolding (S1 Table). Also, two additional systems were used for refolding simulations: i) the isolated CTD of NusG, a protein that shares almost identical secondary and tertiary structure but lacks any observable metamorphic feature (S6 Fig and S1 Table), and ii) a chimeric protein connecting the NTD of RfaH with the CTD of NusG (S5 Fig and S1 Table), in which it is expected that no specific interdomain interactions are formed given the divergent evolution of RfaH and NusG [41]. Remarkably, when the isolated CTD of NusG and its fusion to RfaH NTD were used as input for refolding simulations, the totality of the simulations reached the β-folded state of NusG CTD, regardless of the presence of the NTD (S7 Fig). Although NusG CTD also traverses through a three-strand intermediate state during refolding, it does not become trapped in this configuration as it does the βCTD of RfaH (S6 Fig). Altogether, these data strongly suggest that an interruption in the β-barrel folding process is caused by specific interactions established between RfaH domains.

To gain insights into what interactions are arising between the βCTD of RfaH and its NTD, the majority cluster of the intermediate-folded βRfaH was analyzed in more detail (Fig 5A). A $C_\alpha$ contact map with a threshold of 9.5 Å was calculated for the interaction between the β-intermediate and NTD, as well as αCTD and NTD. In this map, three distinct interaction regions between the βCTD intermediate and the NTD were identified (Fig 5B). Among these, one set comprises native contacts found in the α-fold, corresponding to residues that form the helix $\alpha_2$ of αCTD, or the loop between strands $\beta_3$-$\beta_4$ in the βCTD. Apart from this, the region comprising strand $\beta_1$ (residues 114–123) contains most contacts with the NTD, all of which are absent in the autoinhibited state of RfaH.

To get further information of the nature of the interactions established between the CTD and NTD, we calculated the per-residue tertiary contacts that are minimally or highly frustrated using the protein frustratometer [42]. For this end, the representative structure of the most populated cluster of the intermediate-trapped or completely refolded βCTD, both in isolation and in the context of full-length RfaH, were analyzed using the web version of the protein frustratometer (http://frustratometer.qb.fcen.uba.ar) (Fig 5C and 5D).

When the CTD is successfully refolded, most of the minimally and highly frustrated contacts in the CTD residues are the same throughout this domain, except for residues 123, 145 and 130. Residues 123 and 145 show an increase of more than 10 minimally frustrated contacts when refolded in the full-length RfaH, whereas residue 130 has more minimally frustrated contacts in the isolated CTD. These sets of residues have been identified to be relevant for the stability of the βCTD in previous simulations using dual-basin structure-based models [18] and also for the stability of the autoinhibited state of RfaH in recent NMR experiments of the transformation of RfaH [43]. In contrast, the β-barrel intermediate of the CTD forms more minimally frustrated contacts when in the presence of the NTD than in isolation (Fig 5), particularly doubling the number of these type of contacts in the region corresponding to strand $\beta_1$ and the loop preceding strand $\beta_2$ (residues 114–123). Despite not forming the strand $\beta_1$, such region becomes highly stabilized by bridging interactions between the NTD and the β-barrel intermediate and serves as the interface between the two domains.

The non-native, minimally frustrated interactions that stabilize the β-intermediate in the full-length protein are formed against a hydrophobic patch in the NTD, comprising residues 78–82 and 91–93. It is worth noting that these NTD residues are solvent-protected when RfaH is bound to the TEC [11]. This patch is flanked by a charged and a polar residue, namely H77 and Q95, that are at close distance from two acidic residues of the CTD, E120 and D114. Most of the other CTD residues in between these positions are non-polar and form interactions either with the incipient hydrophobic core of the three-strand intermediate or the

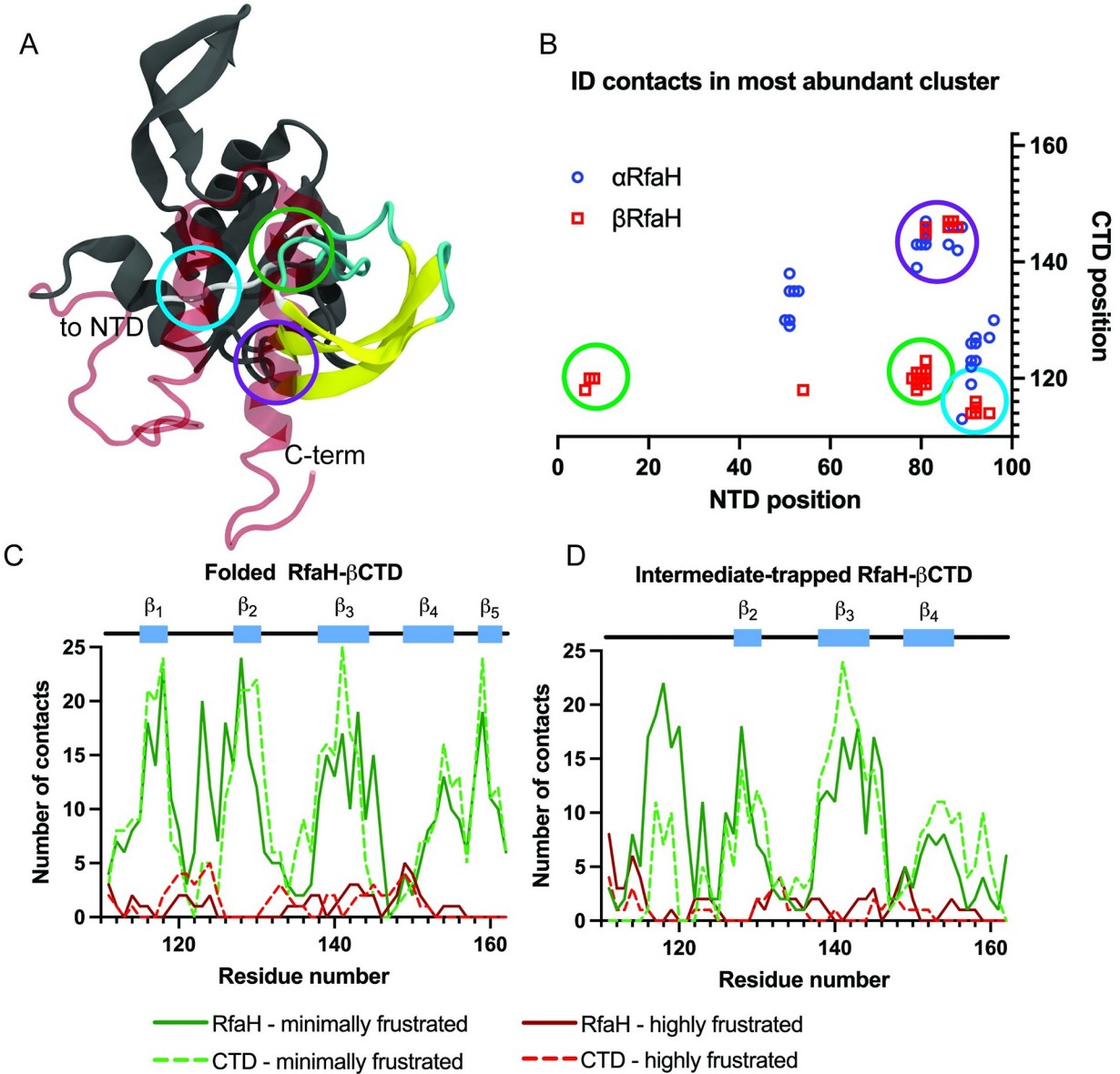

**Fig 5. Contact and frustration analysis of the RfaH β-intermediate and its interaction with the NTD.** (A) Superimposed structures of the αCTD (diffuse, red) and β-intermediate (yellow) on the aligned NTD (gray). The three major points of contacts are circled in different colors. (B) Contact map of the interdomain interface observed in αRfaH (blue) and in the β-intermediate (red). The number of highly (red) and minimally frustrated (green) contacts is shown for the CTD in isolation (dashed line) and in the context of full-length RfaH (solid line) for the completely folded CTD (C) or the β-intermediate (D).

hydrophobic patch of the NTD. Of these residues, the only non-polar residue that does not form part of the hydrophobic core in the folded βCTD is I117.

We also observe a decrease in minimally frustrated contacts in strand $\beta_3$, $\beta_4$ and the C-terminus of the β-intermediate upon binding to the NTD. Upon careful inspection of the contacts taking place in these regions, we noted that the region corresponding to strand $\beta_1$ forms a core of contacts with the C-terminus and the three β-strands in the isolated β-intermediate. This core decreases its amount of intradomain contacts when strand $\beta_1$ encounters the NTD hydrophobic patch rich in minimally frustrated contacts.

## Discussion

*E. coli* RfaH is known as one of the most dramatic examples of protein fold-switching. In solution, RfaH folds into an autoinhibited state in which the αCTD tightly binds to the NTD. This contrasts to the dynamics of its active state, which is only feasible in its full length in the presence of *ops*-paused TEC [43], in which case both domains dissociate and fluctuate independently. In contrast, the non-metamorphic *E. coli* NusG only transiently forms interdomain interactions, existing always in solution as a protein with two independently moving domains [3,5]. Our simulations using the AWSEM MD and force field package correctly model RfaH in all its conformations and recapitulate its thermodynamic behavior in solution, evidenced as the switching of the energetic minimum between αCTD and βCTD when breaking interdomain interactions. This switch has also been observed in previous computational works on full-length RfaH using various simulation strategies [18,37].

More importantly, our refolding simulations show that the number of trajectories that successfully reach the β-folded CTD in the context of full-length RfaH is a minority when compared to the cases in which the CTD becomes trapped in a three-strand β-barrel intermediate, and almost three times less successful than refolding of αRfaH. We also demonstrate that a significant number of minimally frustrated NTD-CTD interactions, some of which are also observed in the autoinhibited state of RfaH, interfere with proper β-fold formation by stabilizing its intermediate state. These results suggest that the thermodynamic stability of the autoinhibited state of RfaH is not only due to the compatibility between the αCTD and NTD but also due to a selective stabilization of the β-intermediate by the NTD, which increases the probability of the β-barrel being trapped in a three β-strands intermediate. Moreover, while refolding of the CTD of the non-metamorphic RfaH paralog NusG successfully reaches the β-folded state, the transient observation of a structurally similar intermediate state also suggests that it is the nature of the NTD and CTD sequence of RfaH that drives the interdomain interaction and ultimate trapping into this state.

Of importance in the refolding process is the configuration that the interdomain linker may take. As it has been previously reported [44], including our own research [38], the linker does play a role in interdomain stability by favoring and stabilizing the αCTD in the hairpin conformation. During our experiments the linker was not given a memory potential, not being stabilized in a particular conformation other than that which arises from the force field for its sequence. We observed the linker to be flexible, not acquiring any degree of secondary structure during our refolding or umbrella sampling simulations. Based on our results and the literature, we hypothesize that αRfaH stabilization by the linker is due to it acting as an entropic spring, i.e., when both domains are close together the linker accesses to a higher number of configurations than when the domains are separated. A similar process may be responsible for allowing the interactions between the β-barrel intermediate and the NTD.

Multiple reports have studied the metamorphic process of RfaH CTD in the context of the isolated domain [14–17] and the full-length protein [18–20], but only a few have described the β-intermediate observed here during βCTD refolding. One of such works corresponds to the computational study of the α-to-β transition of the isolated CTD of RfaH through targeted MD and Markov state models using an adaptive seeding method, in which several en-route ensembles collectively suggests that strands β2, β3 and β4 are relatively stable and form earlier during refolding towards the β-state [14]. Additionally, our previous work with full-length RfaH using dual-basin structure-based models also identified a βCTD-like intermediate that is either free or interacting with the NTD, but with a different topology [18]. Lastly, recent unbiased explicit solvent simulations of the spontaneous α-to-β fold-switch of RfaH CTD using a replica exchange with hybrid tempering method exhibits three-stranded and four-stranded intermediates before reaching the β-folded CTD [45]. Nevertheless, none of these works

described the active role of the NTD in stabilizing such intermediate state nor characterized its role as part of the β-barrel folding process.

We believe that this three-strand intermediate and its NTD-dependent stabilization has been overlooked due to either the granularity of the model used, the absence of sequence-dependent potentials or the velocity with which the system is being driven out of the equilibrium. In fact, the sequence-dependent potential embedded on AWSEM shows its capabilities when simulating the correct refolding of αRfaH to a high fraction of native contacts $Q_W$ even in the absence of knowledge-based contact information of the interdomain interface and the linker connecting both domains, meaning that these simulations are robust enough to discriminate the interactions arising from RfaH sequence in terms of NTD-CTD association. The observation that NusG CTD, unlike RfaH βCTD, is not affected by RfaH NTD in these simulations is confirmation of the latter.

These arguments, alongside the observation of this intermediate in both NusG and RfaH βCTD folding pathways, also suggest that this intermediate is likely a topological solution to the small β-barrel folding process, which could also be necessary for the transition between the α- and β-folds of RfaH. While our previous work using hydrogen-deuterium exchange mass spectrometry show no apparent differences between NusG CTD and RfaH CTD and no indications of intermediate states under native conditions [38], it is possible that the intermediate state observed here requires the addition of chaotropic agents to favor its abundance. It can be presumed that the destabilization of the native state using such approaches not only would favor the intermediate population but also the unfolded state.

All in all, our simulations indicate that the NTD actively participates in thermodynamically favoring the autoinhibited α-state by properly orienting the αCTD and correctly specifying the interactions occurring upon interdomain interface formation and by switching the equilibrium from the β-folded CTD into a folding intermediate. Such intermediate could be potentially observed by studying the equilibrium unfolding of the isolated CTD, as it was observed here during the refolding process of the isolated CTD of RfaH and NusG as well as part of the metamorphic pathway in full-length RfaH. We also hypothesize that stabilization into the β-intermediate by the NTD is the initial step for RfaH to fold-switch back into the autoinhibited state, as the intermediate states observed through umbrella sampling and temperature annealing are structurally the same, i.e., both have three β-strands and share an RMSD value of 2.5 Å (S5 Fig). This idea is compatible with the observation of RfaH stably binding the ribosomal protein S10 through its βCTD when bound to the TEC [43], as in such state the NTD hydrophobic patch is blocked by RNAP. Therefore, the effect of the NTD over the βCTD can only be observed when releasing the active state of RfaH from the TEC, hence the role of the NTD to fold-switch back into the autoinhibited state.

## Supporting information

**S1 Table. Summary of the refolding experiments and features of final refolded states for all systems in this work.**
(XLSX)

**S1 Fig. Interaction matrices for umbrella sampling using $Q_{diff}$.** $C_\alpha$ residue-residue distance matrices for full-length RfaH and its isolated CTD. The matrices grow along the diagonal, which represents the same residue distance, in this case set to 0. Along this diagonal, contacts are formed in a 1–4 residue pattern for α-helices, antiparallel and parallel lines indicating β-strands. The blue blocks indicate regions of high distance (99 Å), which were manually set in order to exclude them from the $Q_{diff}$ calculation.
(TIF)

**S2 Fig. Histograms of the energy and $Q_{diff}$ reaction coordinates in umbrella sampling.** In these umbrella sampling simulations, 51 simulations in $Q_{diff}$ steps of 0.02 were run, totaling 51 simulations per system per temperature. The histograms marked in red were not used for the WHAM analysis as the simulation got trapped in a misfolded configuration. RfaH reaches the α-folded autoinhibited state when $Q_{diff} = 1$ and the isolated CTD reaches the β-folded state when $Q_{diff} = 1$. Although not sufficient sampling was achieved for $Q_{diff} \sim 0.00$ for the full-length protein, the beta configuration was successfully sampled as it is observed in Fig 1B.
(TIF)

**S3 Fig. Heat capacity of RfaH.** Heat capacity calculated from umbrella simulations on the full-length RfaH and the isolated CTD. The blue arrow indicates the temperature selected for presenting the free energy landscape of the isolated CTD in Fig 2A, and the blue arrow indicates the temperature selected for presenting the free energy landscape of the full-length RfaH in Fig 2B. The values on the left y-axis correspond to RfaH, whereas the values on the right y-axis correspond to the isolated CTD.
(TIF)

**S4 Fig. Free-energy landscapes of RfaH over $Q_{diff}$.** The free energy landscapes of isolated CTD (left) or full-length protein (right) were projected onto the $Q_{diff}$ reaction coordinate alone, which describes the transition between α-folded and β-folded CTD.
(TIF)

**S5 Fig. Representative refolding traces for the two-domain constructs used in the work.** The N-C distance shown in green is a measure of how close or separated are the proteins. At low temperatures they tend to agglutinate as a way to minimize the energy, particularly of the exposed NTD hydrophobic patch, which has many residues whose burial energy remains unsatisfied otherwise.
(TIF)

**S6 Fig. The intermediate of the RfaH CTD is the same for NusG CTD.** (A) Annealing plots of RfaH CTD and NusG CTD. Each point was taken every 2,000 steps of $3 \cdot 10^7$ step trajectories that ramped down from 1.6 $T_f$ to 0.6 $T_f$. For both RfaH and NusG, an intermediate is observed at $0.4 \leq Q_W \leq 0.6$. (B) Comparison of refolding traces and intermediate structures of RfaH CTD and NusG CTD. The folding states of both traces was visually inspected. For each trace, the unfolded state is denoted as U, while the intermediate state is denoted as I and the folded state is denoted as F. (C) Structural alignment via STAMP of the intermediate states observed for RfaH and NusG and the RMSD to the folded state for RfaH and NusG. (D) Structural alignment via STAMP of the β-intermediate state observed for RfaH CTD in umbrella sampling and refolding simulations.
(TIF)

**S7 Fig. Refolding of NusG βCTD alone and its fusion to RfaH NTD.** Representative final structures after NusG βCTD refolding in a RfaH NTD–NusG CTD chimera and in isolation. The histograms represent the RMSD distribution of the final structures. All simulations reached the β-folded state of NusG CTD.
(TIF)

## Acknowledgments

The authors thank the generous contribution from Red MacroUniversidades award for graduate mobility to Universidad de Buenos Aires and the gracious help from Franco Tavella during the training of P.G.-D. in AWSEM.

## Author Contributions

**Conceptualization:** Pablo Galaz-Davison, Ernesto A. Román, César A. Ramírez-Sarmiento.

**Formal analysis:** Pablo Galaz-Davison, Ernesto A. Román, César A. Ramírez-Sarmiento.

**Funding acquisition:** Ernesto A. Román, César A. Ramírez-Sarmiento.

**Investigation:** Pablo Galaz-Davison, Ernesto A. Román.

**Methodology:** Pablo Galaz-Davison, Ernesto A. Román, César A. Ramírez-Sarmiento.

**Supervision:** Ernesto A. Román, César A. Ramírez-Sarmiento.

**Visualization:** Pablo Galaz-Davison.

**Writing – original draft:** Pablo Galaz-Davison, Ernesto A. Román, César A. Ramírez-Sarmiento.

**Writing – review & editing:** Pablo Galaz-Davison, Ernesto A. Román, César A. Ramírez-Sarmiento.

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
