## [Decision Letter · Decision Letter 0]

12 Apr 2021

Dear Dr. Ramirez-Sarmiento,

Thank you very much for submitting your manuscript "The N-terminal domain of RfaH plays an active role in protein fold-switching" for consideration at PLOS Computational Biology.

As with all papers reviewed by the journal, your manuscript was reviewed by members of the editorial board and by several independent reviewers. In light of the reviews (below this email), we would like to invite the resubmission of a significantly-revised version that takes into account the reviewers' comments.

We cannot make any decision about publication until we have seen the revised manuscript and your response to the reviewers' comments. Your revised manuscript is also likely to be sent to reviewers for further evaluation.

Sincerely,

Anders Wallqvist

Associate Editor

PLOS Computational Biology

Arne Elofsson

Deputy Editor

PLOS Computational Biology

Reviewer's Responses to Questions

**Comments to the Authors:**

Reviewer #1: RfaH is one of the most extreme examples of a fold-switching protein. The two-domain protein reversibly cycles between two states: in the closed state the C-terminal domain (CTD) folds as a-helical hairpin and binds to the N-terminal domain, rendering RfaH autoinhibited. Upon recruitment to an ops-paused TEC the domains dissociate and the CTD refolds to a NusG-like b-barrel. Although many details of its functional cycle have been deciphered and despite many bioinformatical studies, the molecular basis of CTD refolding is still largely unkown.

In this manuscript the authors use Associative Water-mediated Structure and Energy Model (AWSEM) molecular dynamics applying umbrella sampling and temperature refolding simulations in order to study the transformation of the CTD with a particular focus on how the NTD affects CTD folding. First they confirm the experimentally determined folding states of full-length RfaH and the isolated CTD exploring their conformation space, but, interestingly, their analysis reveals a β-intermediate, which consists of three β -strands and which may be a general folding intermediate in the folding pathway of the β -barrel. The authors show that this intermediate is stabilized by contacts to the NTD and they hypothesize that a destabilization of the β -barrel by the NTD is the first step when RfaH transforms from its activated to the autoinhibited state.

Overall the manuscript is sound and reveals new aspects of the transformation of RfaH.

Comments

1. p. 5, l 9: A general explanation of the “umbrella sampling” should be added so that also non-bioinformaticians can read and understand this section.

2. Fig. 1B: labeling of the points at which the energy landscapes are calculated should be colored according to the curves to facilitate understand; it is also not clear why the letters “C” and “R” are chosen so that arrows colored according to the curves should be sufficient

Fig. 1C: The energy barrier between αRfaH and βRfaH should be marked

3. p.7, l 1-3: Contacts of the flexible linker have been ignored as no defined conformation has been reported, which is reasonable. Nevertheless, the linker might play a role during refolding (either to α or β or both states). This possibility should be included in the discussion.

4. p. 7, l 19-20: The authors should describe the graphs of Fig. S2 and how exactly they relate to the energy landcapes in Fig. 1 (not only technically, i.e. that it’s the 1D projection, but in terms of minima etc). They should also describe how the β-intermediate has been identified/observed (minimum)

As this is the first time that the β-intermediate has been identified, the Fig. S2 should be moved to the main manuscript.

5. p. 8, l 24-25: This kind of behavior is expected for a two-domain proteins the domains of which behave independently not for a two-domain protein in general -> rephrase

6. Fig. 2B and C: to clarify the states they should be labeled “correctly folded domains, correct interface” (B) and “correctly folded domains, incorrect interface” (C); the color code should be explained

7. p. 9, l 10: The refolding simulations of full length RfaH not only recapitulate the structure of RfaH (Ref 8), but also the experimental unfolding/refolding experiments (ref 24)

8. p. 9, l 16-17. It is not clear why solely the NTD-side of the domain interface is responsible for the correct orientation/binding of the CTD; such a conclusion requires a detailed analysis of the contacts and is only valid if specific sidechains of the NTD contact only backbone atoms from the CTD; if also CTD sidechains are involved the specificity relies on both domains

9. Fig. 3 and S3: The labeling “+NTD” and “-NTD” is misleading; it should be “full length RfaH” and “isolated CTD” or “NTD-CTD” and “CTD”

For better comparability the orientation of the CTD should be the same in A and B

Termini should be labeled

The NTD should have a weaker color (e.g. just grey) in order to emphasize what happens with the CTD

10. Fig. 4: labeling: it should be “RfaH – mimimally frustrated”, “RfaH – highly frustrated” etc

The authors state that the β-intermediate forms more minimaly frustrated contacts in the presence of NTD than in the absence. This seems to be true for the overall number of minimally frustrated contacts and for the region preceding strand β2. However, the authors should discuss the fact that there are more minimally frustrated contacts for the isolated CTD than for RfaH especially in strands β3, β4 and the C-terminus.

11. p. 11: Is the three-strand intermediate observed for NusG-CTD the same as for RfaH-βCTD?

12. p. 11: The authors state that NusG CTD also traverses through a three-strand intermediate during refolding. Is this the same intermediate as for RfaH-CTD? Why is this intermediate not shown in Fig. S3?

13. p.13, l 1: the term “exists in solution as two-domain protein” is strange as it was also a two-protein if the domains interacted (tightly) -> needs rephrasing (see #5)

14. p. 13, l9-10: is the intermediate observed in the analysis of the energy landscapes (Fig. S2) the same as that observed during refolding simulations?

15. p. 13, l 10-14: more details about the involved NTD residues should be provided. It should be discussed if these residues are available when RfaH is bound to the TEC. If yes, these contacts would stabilize the intermediate, preventing the CTD to proceed to the full beta-state, and the authors should discuss this

16. p. 13, l15-16: the authors conclude that the NTD actively destabilizes the β -barrel; this has however not been directly demonstrated. It as only been shown in the refolding simulations that the intermediate is stabilized when switching from random coil to beta. Moreover, if the β -state is trapped in the folding intermediate due to stabilizing contacts, what is the driving force to leave the intermediate towards the beta or the α-state (especially to beta as the energy of the intermediate and the beta state seem similar according to Fig. S2)

17. p. 15, l 7-8: The authors hypothesize that the βCTD destabilization by the NTD is the initial step for RfaH refolding into the autoinhibited state. It is not obvious why the NTD actively destabilizes the βCTD. Although there seem to be more minimally frustrated contacts for the intermediate than the bCTD, the b-intermediate seems to be comparable to the β-state energetically (Fig. S2).

18. Discussion: the discussion about the importance of the intermediate may benefit from a scheme that illustrates the refolding steps suggested by the authors (including the most important structures, i.e. autoinhibited, unfolded, intermediate, activated)

19. In general all physical and mathematical quantities/variables/constants (such as “T”, “Qdiff”, “j”, “Q0”) in the text and in graphs (e.g. Fig. 1B) must be in italics!

Minor comments:

p. 3, l 4 a reference should be chosen that reviews the modular structure of NusG proteins, e.g. Werner JMB 2012 or Artsimovitch mbio 2019

p. 5, l 5 and 7: only genes are expressed, not proteins

p. 8, l 21: the chronological order is not correct as Fig. 2B is mentioned before 2A.

Table S1: nomenclature of RfaH is not the same as in the text (αRfaH vs. RfaH-α)

p. 11, l22-26: the sentence is very long and hard to understand and should be rephrased (divided in two or more sentences)

P. 12, l. 17 the reference should be #34, not #13

p.12, l 15-17: The sentence seems not correct

p.13, l 1: ref 5 is not correct and should be replaced by Burmann et al BiochemJ 2011

p. 16, l 22: “diff” must be subscript

Reviewer #2: This work presents the results of folding simulations of the C-terminal domain of the RfaH transcription factor, either isolated or as part of the full-length protein. This domain is known to undergo a transition from a alpha-helical fold to a beta-barrel fold at different stages of its regulation activity. The work appears to highlight the role of interdomain interactions in RfaH in regulating the transformation from one fold to another. However it seems to me that the delivery of the main message of the work, being buried in several layers of other information, could be improved substantially. I am also concerned with several technical aspects of the work as described below. In my opinion, the work in its present form cannot be recommended for publication.

The objectives of the work are very unclear at a structural level. The key description of the system is given in the second paragraph of the Introduction I think. However, personally, I had a very hard time visualizing the main features of the system. A picture/cartoon/diagram could be very useful here to summarize the system and the process. Perhaps the diagram could also pinpoint the interdomain interactions referenced in the following paragraphs that appear to be main subject of study.

I found that, generally, the primary objectives and results of the work are hard to grasp when buried in long discussions interdispersed with acronyms and jargon. I recommend that these are condensed in a brief sentence. What does this work add to the extensive previous modeling studies of this system?

The presentation of the Methods after the Results does not help the presentation. The Results refer to quantities (Qdiff, Qw, etc.) defined in the Methods whose significance is unclear to reader. Sentences such as "RfaH interdomain contacts were kept as bias for simulating the αRfaH state in the full-length system, while all contacts between the βCTD and NTD in the cryo-EM structure were ignored by increasing the distance in the residue-residue distance matrix beyond the threshold from which a contact is considered to take place (9.5 Å)." is very confusing to the reader if terms such as "bias", "distance matrix", "threshold", are not described, at least qualitatively.

The AWSEM model comes out a bit from the blue. It would be useful to discuss early one of why it was adopted and why it is believed to be a suitable model for this system. I could not understand what the "memory" potential is, and what is the significance of "single" or "dual" memories.

Qdiff page 16, should probably be (q-qB)/(qA-qB) so that Qdiff goes from 0 to 1 as the structure goes from B to A as suggested by the language of the paragraph following the third equation on this page. As defined Qdiff's range in not [0,1].

In the following equation, what is the meaning of rij raised to NA or NB? NA and NB are not defined but the notation suggests that they are the number of residues of the two reference structures? Or are NA and NB simply labeling the two reference structures? If the latter, why not use simply A and B?

This equation is also dimensionally ill-defined. The exponents should be dimensionless but the units of distance square in the numerator do not match the units of sigmaij, which is dimensionless.

Page 17. Here and elsewhere, supporting data in the the supplementary information are referenced as results. For example "secondary structure assignments are summarized in Table S1 ..." In my opinion, if they are so important to be referenced in the main text, supporting data should be shown in the main text. Otherwise, references should be omitted and the content of the supplementary information should be summarized elsewhere so as not to confuse the reader of which results are essential to the work and which ones are available for confirmation, if needed.

Reviewer #3: Galaz-Davison et al. describes computer simulations of RfaH, a protein that undergoes so-called fold switching, i.e., a reversible structural transformation from one fold to another. The focus in this work is on the role of the N-terminal domain (NTD) of RfaH for this transition. Fold switching as a phenomenon is becoming increasingly recognized as an important mechanism in both protein function and evolution and RfaH is one of the most well studied fold switching proteins, both experimentally and theoretically. Depite this, there has been very little attention given to role of the NTD (the part of RfaH left unchanged during fold switching) in previous studies, making the present study timely.

A major strength of this work is the folding “annealing” simulations carried out on RfaH and the related protein NusG. NusG is of interest because it is a homolog of RfaH but does not exhibit fold switching. It is shown that for RfaH the folding of the C terminal domain (CTD) into its beta fold is hampered by energetically favorable interactions with the NTD. However, these interactions are not as strong for NusG. For NusG, folding of the CTD proceeds undisturbed even in the presence of the NTD. This is an important results because it hints at the mechanism of the “reverse” fold switch (i.e. how the betaCTD fold reverts back to its alpha fold, which is neccessary to reset RfaH to its autoinhibited state). Overall, the present study is of significant interest for experimental and theoretical researchers in the field of protein fold switching. The manuscript is generally clear and well written (although see points below), and includes a very nice Discussion section.

With the above in mind, I have a few questions and concerns regarding some of the methodology and analysis. These questions should be addressed before the manuscript can be accepted for publication.

One of my main concerns is with the umbrella sampling strategy that are used to obtain the equilibrium behavior both the complete RfaH protein and the C terminal domain (CTD) in isolation. The AWSEM model (used for all simualtions) includes a term V_memory, which is based on information from experimental structures of RfaH. However, it is not clear in the manuscript how V_memory is applied. Page 16 states generally that V_memory “can be guided to multiple structures, or as used in this work, limited to a single reference structure” but further down on the same page “dual memories” are mentioned in relation to the choice of initial structures. How V_memory is incorporated into to the umbrella sampling simulations should be clarified. Since V_memory is central to the present study, I suggest also including sore more general information about this term, e.g., functional form, free parameters, to inform the reader.

In the umbrella sampling simulations, the AWSEM model is combined with a harmonic potential in the parameter Q_diff. Does this mean that there are two different types of biases in these simulations (V_memory and the umbrella bias)? Are both types of biases needed to capture fold switching? It would be interesting to see some simulation results of AWSEM without the umbrella sampling, as a point of comparison. Typically, umbrella sampling is seen merely a trick to enhance sampling of conformational space and should in principle not influence the resulting (equilibrium) behavior of the underlying model. Or should the umbrella weight function be seen here as an additional “effective” bias towards the two folds? It would be useful if this issue could be clarified.

I also have some concerns with the analysis of the umbrella sampling simulations. It seems somewhat optimistic that the results from a single temperature (please specify which T in the text) can be accurately reweighted to span the temperature range ~500-800 Kelvin using WHAM, especially if the simululations were carried out below the folding temperature. It is possible that the simulations sufficiently well samples the unfolded state, since they span states in between the alpha and beta folds. However, this can not be assumed a priori. Could a few separate simulations at higher temperatures be carried out to confirm the results from the WHAM analysis?

Some clarifications are needed regarding the definition of order parameters Qw, Qdiff and q. i) According to page 17, Qw is calculated by summing over all (N-2)(N-3)/2 residue-residue pairs j>i+2. But on page 6, it is mentioned that beta CTD-NTD contacts are ignored by increasing the distance in the residue-residue distance matrix beyond 9.5 Å. ii) There appears to be a sign error in way Qdiff is defined. Inserting q_A for q leads to Qdiff = 0 and inserting q_B for q gives Qdiff=-1, but the range should presumably be 0 to 1. iii) Is it possible to define q in terms of Qw, thereby simplifying expression for q? iv) The width of the gaussian functions is not constant but taken to be sigma = |i-j|^0.15? Please comment on this choice.

Minor points:

1) How was the value of lambda_FM=0.30 determined? Presumably, this parameter controls the strength of the fragment memory term (please clarity). Why does a higher lambda_FM lead to a stronger cooperativity? These points should be clarified. 

2) Page 5 line 21. “Just before” should presumably be “just below”?

3) Page 7, line 7. The free energy surface in Fig. 1C is described as having a “single, deep energy minimum”. I suggest rephrasing this sentence as any protein energy landscape will have a multitude of minor minima throughout its energy landscape.

4) Page 16. It would be helpful if the basic idea of umbrella sampling is explained in a few sentences at the beginning of the section “Qdiff and umbrella sampling”.

5) The histograms in Fig. S1 indicate appropriate overlaps between consecutive Qdiff and energy distributions, however, they are hard to read. Perhaps it would be worth trying lines rather than box histograms.

7) Fig. S4. Refolding “curves” is probably not a good description of these scatter plots.

8) It would be nice to see some actual (re)folding trajectories, showing, e.g., Qw or RMSD values as a function of simulation time.

9) Please include figures showing the full residue-residue distance matrix referenced on page 6. Alternatively, if Qw, q, etc, are determined using sets of native contacts, please report all contact maps, number of contacts in each set, and the criterion for a contact.

Reviewer #4: One of the domains of the protein RfaH folds to a β-rich fold in isolation but switches to an α-rich fold in the presence of its other domain. RfaH has been used as a model protein to understand the molecular basis of fold switching. In this manuscript, the authors simulate RfaH in order to understand the mechanism of fold switching and the effect of the non-switching domain on this mechanism.

Comments:

(1) Since RfaH has been simulated with many models (including the authors’ previous work), it would be useful to have a detailed discussion up front of what each of the models include/exclude in terms of force field and what is gained from using a specific model. This should help to focus on the benefits of using AWSEM as against other models.

(2) The AWSEM force field has several terms and additionally can be used with or without native structure bias. For readers who have not seen AWSEM before, it would be useful to discuss each of the terms, clearly specifying which of the terms have a native-bias in each of the simulations. This is important for fully grasping the results.

(3) It would be useful to have the same tics and tic labels on Figs. 1C and 1D for ease of visual comparison. Is the third minimum (12,16) in Fig. 1D the unfolded ensemble? What is at (5,5) in 1C and why is it absent in 1D? And where would the beta-barrel intermediate lie on these landscapes? The different basins on this figure should be labelled. The landscapes should be re-labelled free-energy landscapes and free-energy and energy should not be used interchangeably. The temperature units should be explained or reduced units should be used since the units are far from real temperatures.

(4) The authors state: “One disadvantage of the umbrella sampling simulations is that, by directly employing the number of native contacts of the system in αRfaH and βRfaH as collective variables to drive the structural interconversion of RfaH, the formation or disruption of interdomain contacts between specific residue pairs is also biased.” on page 8. Why does unbiasing not work for this? What else is affected by the umbrella sampling which cannot be unbiased?

(5) The authors perform a frustratometer analysis of the intermediate with full length RfaH. However, they do not indicate if the extra minimally frustrated contacts that they see are native contacts seen at the αCTD-NTD interface or are these non-native contacts. Also, if these contacts are native-like then are they between the hydrophobic patch which is present in the NTD of RfaH and absent in NusG. There should be sequence signatures present in the simulations and the analysis performed here which should allow the authors to predict one or a few mutations which destabilize the fold switching or change its kinetics. The authors should make such predictions. They should also be able to redo the frustratometer analysis using the intermediate structures but with mutated residues, as a first test for their mutations. Such mutations and tests will add to the strength of this manuscript.

**Have the authors made all data and (if applicable) computational code underlying the findings in their manuscript fully available?**

Reviewer #1: Yes

Reviewer #2: **No: **I am not sure. But the authors appear to state that they will make the data available on their website after publication.

Reviewer #3: Yes

Reviewer #4: Yes

PLOS authors have the option to publish the peer review history of their article (what does this mean?). If published, this will include your full peer review and any attached files.

Reviewer #1: No

Reviewer #2: No

Reviewer #3: **Yes: **Stefan Wallin

Reviewer #4: No
---

## [Decision Letter · Decision Letter 1]

10 Jul 2021

Dear Dr. Ramirez-Sarmiento,

Thank you very much for submitting your manuscript "The N-terminal domain of RfaH plays an active role in protein fold-switching" for consideration at PLOS Computational Biology. As with all papers reviewed by the journal, your manuscript was reviewed by members of the editorial board and by several independent reviewers. The reviewers appreciated the attention to an important topic. Based on the reviews, we are likely to accept this manuscript for publication, providing that you modify the manuscript according to the review recommendations.

Sincerely,

Anders Wallqvist

Associate Editor

PLOS Computational Biology

Arne Elofsson

Deputy Editor

PLOS Computational Biology

[LINK]

Reviewer's Responses to Questions

**Comments to the Authors:**

Reviewer #2: The revised methods section is substantially improved relative to the original manuscript. However, at least two of the equations (the equations are not numbered in the manuscript) are still problematic. Terms with dimension of distance squared cannot be present as exponents. In my opinion, the manuscript cannot be published unless these are corrected.

Reviewer #3: The authors have clarified basically all the issues that I raised in my comments, including the functional form of V_memory and the structural parameters (q, Qdiff, etc). Regarding the WHAM analysis, it would have been nice to see a comparison of separate analyses of the two sets of simulations carried out (the results should in principle agree). However, it appears that the WHAM analysis of the combined simulations as presented in the revised manuscript (Fig. 2) are in qualitatively in agreement with those of the original submission, indicating agreement.

From my perspective, the manuscript can be accepted for publication if the following (minor) points are addressed:

1) The description of the umbrella sampling simulations in Methods should be updated to include information about the two different temperatures used.

2) It would help the reader if Tf is explained in Results upon first usage in this section.

3) Expressing temperatures in units of Tf is probably a good idea. However, I suggest reporting also the nominal value of Tf in AWSEM units, such that the two scales can be linked. This will be helpful in case someone would like to perform similar types of simulations in the future.

Reviewer #4: The manuscript is mostly okay in its current form. I have three minor comments:

1) Is the memory of the entire RfaH-NTD-alpha-CTD structure encoded in any of the simulations (in particular the refolding simulations)? Or are there only memories of the NTD and the two conformations of the CTD separately? Specifically, are the interdomain interactions between the RfaH NTD and RfaH CTD purely due to the transferable parts of the AWSEM potential? This should be clarified in the manuscript. A similar clarification should be given for the hybrid NusG NTD-RfaH-CTD model.

2)To me, this fold switch looks similar to most ligand induced conformational transitions: the beta-RfaH-CTD is the “open” structure because it has a larger number of stabilizing contacts than alpha-RfaH-CTD. Ligand binding, here binding to the NTD, increases the stabilization of the alpha-RfaH-CTD and this allows the conformational change. If the authors agree, they should put this fold switch in the context of ligand induced conformational transitions in general.

3) The manuscript needs to be copyedited to remove some non-standard phrasing and sentence construction.

**Have the authors made all data and (if applicable) computational code underlying the findings in their manuscript fully available?**

Reviewer #2: Yes

Reviewer #3: Yes

Reviewer #4: None

PLOS authors have the option to publish the peer review history of their article (what does this mean?). If published, this will include your full peer review and any attached files.

Reviewer #2: No

Reviewer #3: No

Reviewer #4: No

Figure Files:

Data Requirements:

Reproducibility:

References:

---

## [Editor Report · Decision Letter 2]

7 Aug 2021

Dear Dr. Ramirez-Sarmiento,

We are pleased to inform you that your manuscript 'The N-terminal domain of RfaH plays an active role in protein fold-switching' has been provisionally accepted for publication in PLOS Computational Biology.

Best regards,

Anders Wallqvist

Associate Editor

PLOS Computational Biology

Arne Elofsson

Deputy Editor

PLOS Computational Biology

---

## [Editor Report · Acceptance letter]

27 Aug 2021

PCOMPBIOL-D-21-00427R2 

The N-terminal domain of RfaH plays an active role in protein fold-switching

Dear Dr Ramirez-Sarmiento,

I am pleased to inform you that your manuscript has been formally accepted for publication in PLOS Computational Biology. Your manuscript is now with our production department and you will be notified of the publication date in due course.

With kind regards,

Andrea Szabo
